# Fatty Acid-Binding Proteins: Their Roles in Ischemic Stroke and Potential as Drug Targets

**DOI:** 10.3390/ijms23179648

**Published:** 2022-08-25

**Authors:** Qingyun Guo, Ichiro Kawahata, An Cheng, Wenbin Jia, Haoyang Wang, Kohji Fukunaga

**Affiliations:** 1Key Laboratory of Brain Science Research & Transformation in Tropical Environment of Hainan Province, Hainan Medical University, Haikou 571199, China; 2Department of CNS Drug Innovation, Graduate School of Pharmaceutical Sciences, Tohoku University, Sendai 980-8578, Japan; 3BRI Pharma Incorporated, Sendai 982-0804, Japan

**Keywords:** ischemic stroke, ischemic cascade, fatty acid-binding protein, neurovascular unit, mitochondria

## Abstract

Stroke is among the leading causes of death and disability worldwide. However, despite long-term research yielding numerous candidate neuroprotective drugs, there remains a lack of effective neuroprotective therapies for ischemic stroke patients. Among the factors contributing to this deficiency could be that single-target therapy is insufficient in addressing the complex and extensive mechanistic basis of ischemic brain injury. In this context, lipids serve as an essential component of multiple biological processes and play important roles in the pathogenesis of numerous common neurological diseases. Moreover, in recent years, fatty acid-binding proteins (FABPs), a family of lipid chaperone proteins, have been discovered to be involved in the onset or development of several neurodegenerative diseases, including Alzheimer’s and Parkinson’s disease. However, comparatively little attention has focused on the roles played by FABPs in ischemic stroke. We have recently demonstrated that neural tissue-associated FABPs are involved in the pathological mechanism of ischemic brain injury in mice. Here, we review the literature published in the past decade that has reported on the associations between FABPs and ischemia and summarize the relevant regulatory mechanisms of FABPs implicated in ischemic injury. We also propose candidate FABPs that could serve as potential therapeutic targets for ischemic stroke.

## 1. Introduction

Stroke is the second leading cause of death worldwide, and, according to statistics collected in 2019, there were annually 12.2 million incident cases of stroke, 101 million epidemic cases, and 6.55 million stroke-related deaths [1], which, not surprisingly, poses a major challenge to the healthcare systems of different countries. The lifetime risk of stroke from the age of 25 years is 24.9%, with geographical differences, although no significant differences between genders [2]. Moreover, approximately 85% of strokes are ischemic strokes caused by vascular occlusion, of which approximately 75% are attributable to embolism, the most common cause of regional cerebral blood flow obstruction [3]. Although early blood flow restoration mediated via thrombolysis or mechanical thrombectomy is the preferred treatment to limit post-stroke brain damage [4,5], brain tissue damage is caused not only by ischemic injury, but also by oxidative stress and inflammation following reperfusion, which are sources of additional damage to the cerebral microcirculation and adjacent brain tissue, a condition known as cerebral ischemia/reperfusion injury (CI/RI) [3,6]. In the area of ischemic lesion, all cells of the neurovascular unit are injured, resulting in a series of complex ischemic cascade reactions [7]. However, the routinely administered neuroprotective drugs are difficult to break down separately, thereby highlighting the necessity to develop neuroprotective therapies based on the identification of core targets engaged in multiple damage processes. As intracellular lipid chaperones, fatty acid-binding proteins (FABPs) are involved in diverse aspects of cellular function and viability and are also closely linked to neurodegenerative diseases [8,9,10]. In this review, we summarize the current trends in FABPs and related injury mechanisms in clinical studies and animal models of ischemic stroke, and further examine the relationship between FABPs and ischemic brain injury, as well as the potential utility of these proteins as targets for pharmaceutical intervention.

## 2. Pathophysiology of Ischemic Stroke

CI/RI is defined as a damage caused by the rapid restoration of blood supply to brain tissue after a period of ischemia and is a common feature of ischemic stroke [11]. It is thought to play a major role in the pathophysiology of acute ischemic stroke, because secondary brain injury owing to abrupt reperfusion has been shown in preclinical studies to contribute to 70% of the final ischemic lesions [12]. Spontaneous reperfusion occurs in approximately 50–70% of ischemic stroke patients [13]. There is no doubt that CI/RI has become a serious problem, leading to a poor prognosis in stroke patients [14,15]. CI/RI is a multi-faceted and complex pathophysiological process involving multiple interrelated events, including energy exhaustion, intracellular calcium overload, excitotoxicity, mitochondrial dysfunction, oxidative stress, inflammation, microglial activation, blood–brain barrier (BBB) disruption, and leukocyte infiltration [13,16]. These factors interact to form a complex regulatory network, the development of which is closely associated with the severity of ischemia and the time of reperfusion, and which involves a range of cell types that form a series of ischemic cascades that directly or indirectly lead to neuronal apoptosis, death, and necrosis. Untreated patients with ischemic stroke lose an average of 2.03 million neurons, 14.8 billion synapses, and 12.8 km of myelinated fibers per minute [17].

CI/RI typically involves an interaction between oxidative stress and inflammation and represents the basis of the development of the ischemic cascade [18]. Oxidative stress is caused by the excess production of reactive oxygen species (ROS), such as superoxide anion radicals, hydroxyl radicals, and nitric oxide, during cerebral ischemic events, which can contribute to lipid peroxidation, inflammation, and apoptosis. During the reperfusion phase, this stress can be exacerbated as the sudden restoration of oxygenated blood triggers further ROS production [6,19]. Furthermore, CI/RI also has the effect of altering the balance between pro- and anti-inflammatory factors, thereby resulting in the accumulation of a large number of pro-inflammatory cytokines in the center and periphery of ischemic lesions, thus stimulating or aggravating the inflammatory response. The activation and infiltration of inflammatory cells and the synthesis and secretion of adhesion molecules are cascade processes that promote one another [18]. Furthermore, ischemic lesions in the penumbra, which is the key focus for neuroprotective therapy, are characterized by the invasion of inflammatory cells (activated microglia, blood-derived macrophages, neutrophils, and T lymphocytes) [20]. In addition, the release of the inducible matrix metalloproteinases (MMPs) MMP-3 and MMP-9 from microglia, macrophages, and infiltrating neutrophils contributes to the degradation of the extracellular matrix and tight junction proteins, thereby leading to BBB disruption [21], the most serious consequence of which is hemorrhagic transformation in ischemic areas [22]. 

These cascades are present in all cells of the neurovascular unit components, and the complexity of intercellular signaling between components that impairs or promotes neuronal viability cannot be overemphasized. Over the past few decades, cytoprotective monotherapy studies have tended to focus primarily on drugs that target just one component of the ischemic cascade [7]. In general, however, this single-target therapeutic approach has limited impact on the complex set of pathophysiological mechanisms triggered by CI/RI. Consequently, examining the potential utility of multi-target neurovascular unit protection strategies in response to the ischemic cascade is considered to represent a more profitable direction for developing neuroprotective strategies for the treatment of ischemic stroke [23].

## 3. Fatty Acid-Binding Proteins

### 3.1. Expression and Function of FABPs

FABPs are a family of intracellular fatty acid (FA) transport-related proteins that reversibly bind to hydrophobic ligands with different affinities, including saturated FAs, monounsaturated FAs (MUFAs), polyunsaturated FAs (PUFAs), eicosanoids and other lipids, and play important roles in lipid metabolism, gene regulation, and signal transduction [24,25]. They are a low molecular weight intracellular protein with a molecular weight of 14 or 15 kDa that occur as 10 isoforms in humans, and are classified according to the organ or tissue in which they were originally identified and are most predominant [26,27]: L-FABP/FABP1 (liver type), I-FABP/FABP2 (intestinal), H-FABP/FABP3 (cardiac), A-FABP/FABP4 (adipocyte), E-FABP/FABP5 (epidermal), IL-FABP/FABP6 (ileal), B-FABP/FABP7 (brain), M-FABP/FABP8 (myelin), T-FABP/FABP9 (testis), and R-FABP/FABP12 (retinal).

Of the aforementioned 10 isoforms, only FABP3, FABP5 and FABP7 are typically expressed in the neural tissue of the brain [28,29]. Among these, FABP3 is expressed exclusively in neurons, and although scarcely detectable in the brain during embryonic development, its expression gradually increases after birth until adulthood [30]. FABP3 has been found to have a high affinity for n-6 PUFAs and is believed to assist in the consolidation and maintenance of the differentiated state of neurons in the adult brain by utilizing PUFAs [27]. The lack of the FABP3 gene reduces arachidonic acid uptake, and the quality and composition of phospholipids in the mouse brain [31] and can also regulate dopamine signaling by interacting with dopamine D2 receptors [32]. FABP5 has a high affinity for n-3 and n-6 PUFAs and is expressed most extensively in the epidermis, liver, and adipocytes. In contrast to FABP3, FABP5 is highly expressed in the neuronal and glial cells of the embryonic brain, whereas its expression steadily declines after birth and is only weakly detected in the adult brain [29]. Moreover, FABP5 has been established to control the regulation of developmental genes expression programs activated by lipid messengers such as retinoic acid (RA), and effectively promotes cell proliferation by activating peroxisome proliferator-activated receptor δ (PPARδ) [33]. It has been proposed that FABP5 may contribute to neurite outgrowth and axonal development by providing long-chain FAs (LCFAs) for phospholipid synthesis [27]. FABP5 also plays roles in neurogenesis, neuronal migration, neuronal terminal differentiation, and astrocyte development, and is required for neuronal development. Similar to FABP5., FABP7 has been established to have a high affinity for n-3 PUFAs, although is mainly expressed in astrocytes and oligodendrocyte cells. Additionally, like FABP5, FABP7 is expressed at high levels in the embryonic brain and at low levels in the mature brain and is essential for the maintenance and proliferation of neural stem progenitor cells and radial glial cells [29,33].

Among the other FABPs, FABP1 is abundantly present in the cytoplasm of hepatocytes, but also expressed in enterocytes and renal tubular cells. As an important endogenous cytoprotective agent, FABP1 reduces oxidative damage to hepatocytes and agonists in liver I/R and other injuries [34], whereas the downregulation of FABP1 expression protects against non-alcoholic fatty liver disease [35,36]. FABP2 is similarly found in enterocytes and has been established to play roles in the absorption of dietary fatty acids in the gut [37]. Its content in the systemic circulation serves as a biomarker for human intestinal disorders such as necrotizing enterocolitis and ileitis [38]. FABP6 is also present in the distal region of the small intestine, wherein it functions as an intracellular transporter of bile acids in ileal epithelial cells, assisting in the catalysis and metabolism of cholesterol [39]. FABP4 is mainly expressed in differentiated adipocytes and macrophages, in which it is involved in metabolic and inflammatory pathways, contributing to the development of insulin resistance, diabetes and atherosclerosis [40]. FABP8 is found abundantly in the myelin sheaths and Schwann cells of peripheral nerves and has been found to function in tandem with myelin basic protein to stack phospholipid membranes and exert unique functions in the organization and stabilization of myelin multilayers [27]. FABP9 maintains sperm quality [41]. Although little is currently known regarding FABP12, the most recently discovered member of the FABP family [26], this protein is assumed to play protective roles by targeting and activating PPAR/RXR in the retinal ganglion cells [42].

### 3.2. Functions of FABP3, 5, and 7 in Neurodegeneration

The three FABPs expressed in the neural tissues of the brain have been implicated in the clinical and pathological manifestations associated with the progressive loss of neuronal structure or function, collectively described as neurodegenerative diseases [43]. α-Synuclein (αSyn) has been established as a pathological hallmark of synucleinopathies, and in conjunction with fibrillation, the misfolding of this protein triggers the occurrence of Parkinson’s disease (PD), dementia with Lewy bodies (DLB), and multiple system atrophy (MSA) [44]. The findings of clinical studies have indicated that FABP3, which is expressed in mature neurons, colocalizes with αSyn aggregates in the brains of PD and MSA patients, although not with amyloid β or phospho-tau aggregates in the brains of individuals suffering from Alzheimer’s disease (AD) [8]. Previous studies have also revealed that FABP3 promotes the oligomerization of αSyn in the brain of a 1-methyl-4-phenyl-1,2,3,6-tetrahydropyridine (MPTP) mouse model of PD [45], and that the deletion of FABP3 can block the uptake of αSyn by dopaminergic neurons and even the dispersion of αSyn in the brain [46]. Given that the levels of FABP3 are significantly higher in the cerebrospinal fluid and serum of patients with AD, DLB, PD with dementia, and vascular dementia (VaD) than in healthy individuals, it has been proposed as a biomarker of neurodegeneration [47,48,49,50], with high levels of FABP3 being significantly associated with the development of dementia [10]. 

The expression of FABP5 is similarly upregulated under pathological conditions, as observed in the dorsal root ganglia neurons of rats with peripheral axonal injury [27] and the postmortem cerebral cortex of schizophrenic patients [51]. Moreover, FABP5 has been established to have a close functional relationship with αSyn, with which it binds to form a large molecular weight complex in Neuro-2A cells treated with rotenone (a mitochondrial complex I inhibitor) and triggers a reduction in mitochondrial membrane potential and loss of cell viability [52]. Abnormal accumulation of FABP5 has been observed in the in mitochondria of psychotropic drug-treated oligodendrocytes, and FABP5 has been shown to co-form mitochondrial macropores with voltage-dependent anion channel-1 (VDAC-1) and Bcl-2-associated X protein (BAX) [53,54]. Furthermore, mice in which FABP5 has been knocked out have been found to be characterized by learning and memory deficits, which are associated with reductions in the activation of PPARβ/δ by arachidonic acid [55]. Additionally, in dendritic cells and T lymphocytes, FABP5 has been demonstrated to promote a hyperactive immune response in an experimental autoimmune encephalomyelitis (EAE) mouse model of multiple sclerosis [56,57]. 

The findings of several studies have provided evidence that FABP7 is also implicated in pathological processes in the central nervous system. For example, the levels of FABP7 levels in the serum of patients with AD, PD, and other neurological disorders associated with dementia are considerably higher than those in non-dementia patients [58]. It has also been found that compared with healthy controls, levels of FABP7 mRNA are higher in the postmortem brains of patients with autism spectrum disorder [51] and schizophrenia [59]. Simultaneous allelic association studies have also revealed that the expression of FABP7 is associated with schizophrenia and bipolar disorder [60]. Moreover, following EAE induction, elevated levels of FABP7 are observed in glial fibrillary acidic protein (GFAP)-positive astrocytes in the demyelinated region in WT mice, whereas, although FABP7 deletion accelerates the early onset of EAE-induced spinal cord inflammation, FABP7 knockout mice exhibit less chronic EAE symptoms. [61,62]. Additionally, the levels of FABP7 are also significantly upregulated in the occipital cortex of Down’s syndrome patients [63].

### 3.3. Changes in FABPs following Ischemic Events

Over the past 10 years, several researchers have focused on the change trends of FABPs following ischemic injury to the brain, tissues, or organs, the findings of which are summarized in Table 1. These studies indicate that the levels of FABP1–FABP7 are all upregulated in response to ischemic episodes and that these FABPs are associated with ischemia in the tissues in which they are highly expressed. For example, elevated levels of FABP1 are recorded in the urine of patients with renal I/R injury [64], whereas significantly elevated levels of FABP2 are detected in patients with intestinal infarction [65,66]. Similarly, higher levels of serum FABP3 are observed in patients with ischemic stroke [67] or myocardial infarction [68], and elevated levels of FABP7 are recorded in the serum of acute ischemic stroke patients [67]. Despite a lack of relevant mechanistic data in these clinical trials, these observations tend to indicate that FABPs have potential utility as biomarkers after ischemic injury based on the analysis of clinical relevance. Furthermore, among the currently identified FABPs, only FABP3, FABP4, FABP5, and FABP7 have been established to be involved in ischemic stroke, and, as yet, there have been no reports to indicate that FABP8, FABP9, and FABP12 are associated with ischemic events, which may be ascribed to their high tissue specificity. According to data obtained to date, the functions of FABPs in ischemic brain injury are closely associated with lipid peroxidation, pro-apoptotic signaling. and inflammation. 

## 4. Changes in FABPs in Ischemic Stroke

At present, we have a preliminary understanding of the changes and general mechanisms of FABPs that occur during ischemic events, and, here, we further focus on the associations between FABPs and ischemic stroke. To clarify the differential expression of FABPs in patient samples and animal models of ischemic stroke, we collected an mRNA sequence dataset of blood obtained from cardioembolic stroke patients (accession number: GSE58294), as well as mRNA sequence datasets of blood (GSE119121) and brain tissue (GSE97537) samples from rats subjected to modeled middle cerebral artery occlusion (MCAO) ischemic stroke. These were obtained from the publicly available Gene Expression Omnibus (GEO) database at the National Center for Biotechnology Information (NCBI), using which, we analyzed differential gene expression. The Venn diagram shown in Figure 1 revealed that the expression of 2454 genes was significantly altered in both clinical samples and animal model samples, among which, FABP3, FABP4, FABP5, and FABP7 are the FABPs with significantly altered expression profiles (Figure 1A). These findings are consistent with those of the previous studies summarized in Table 1.

Having established the changes in the expression of FABP genes in the datasets, we proceeded to visualize these changes using volcano plots. The mRNA expression levels of FABP3-FABP7, and, in particular, FABP4 and FABP5, were shown to be significantly higher in the blood samples obtained from both patients (Figure 1B) and MCAO rats (Figure 1C). However, in the brain tissue samples of MCAO rats (Figure 1D), only FABP4, FABP5, and FABP7 were upregulated, whereas FABP3 was downregulated. Furthermore, a comparison of the changes in FABP expression levels in blood samples (Figure 1E) and the brain tissues (Figure 1F) of MCAO mice revealed that the change trends of FABP4, FABP5, and FABP7 were synchronous and obvious. Contrastingly, whereas we detected the elevated expression of FABP1 and FABP6 mRNA in blood samples, we failed to observe similar changes in brain tissue, thereby highlighting the tissue specificity of these isoforms. Although it remains to be determined how ischemic brain injury promotes the release of FABP1, which is strongly expressed in the kidney, and FABP6, which is highly expressed in the ileum, into the circulatory system, at least these two FABPs should not be directly related to ischemic injury of brain tissue. This finding also raises the question as to how FABP1 and FABP6 detect signals associated with nervous system damage and how they respond on receiving this information. Once we succeed in identifying the implicated regulatory pathways, it may then be possible to determine how ischemic brain injury leads to distal organ dysfunction or damage [94,95].

In addition, the findings of our previous quantitative analyses of FABP3, FABP5, and FABP7 proteins in the brain tissues of MCAO mice [78] have revealed that the levels of FABP5 and FABP7 protein increased with prolonged duration of reperfusion (Figure 1G–I), which is consistent with the results obtained from the analysis of the GSE97537 dataset. However, we additionally found that the level of FABP3 protein expression was also significantly increased, although not as high as that of FABP5 and FABP7. We speculate that the disparity between the results presented in Figure 1D,G could be explained in terms of differences in the animals examined and the duration of ischemia. On the basis of these data, we therefore hypothesize that FABP3, FABP4, FABP5, and FABP7 are directly implicated in acute ischemic stroke, and that the latter three isoforms appear to be more active in responding to the onset of ischemia.

## 5. The Role of FABPs in the Ischemic Cascade

Since their discovery, numerous functions have been proposed for FABPs. As lipid chaperones, these proteins can actively promote lipid transport to diverse cellular compartments and are involved in lipid droplet formation, lipid oxidation, membrane synthesis, signaling, and the nuclear regulation of transcription, all of which are essential for maintaining the functional homeostasis of cells [24]. For example, by interacting with dopamine D2 receptors, FABP3 has been demonstrated to be involved in the release of acetylcholine and glutamate in the mature mouse brain [32], whereas FABP5 has been established to regulate learning and memory in mice by reducing anandamide levels and activating the nuclear receptor PPARβ/δ [55]. Furthermore, FABP7 has been demonstrated to play protective roles against spinal cord compression injury in mice [96]. Unexpectedly, however, the activities of FABPs following an ischemic event appear to be detrimental with respect to neuroprotection. The following is a synopsis of the mechanisms whereby selected FABP subtypes participate in the ischemic cascade.

### 5.1. FABP3

The expression of FABP3 protein has been observed to be significantly increased in the ischemic penumbra neurons of MCAO mice [78], thereby implying that this protein may be directly associated with the survival or apoptosis of ischemic neurons. By up-regulating apoptotic signals and the phosphorylation of mitogen-activated protein kinase (MAPK) signaling pathways, the overexpression of FABP3 has been shown to promote myocardial cell death in a myocardial infarction mouse model, whereas deficiency in FABP3 can effectively prevent ischemic heart injury [79]. The overexpression of FABP3 has also been observed to reduce cellular ATP production, concomitant with an apparent reduction in mitochondrial membrane potential, and promotes excessive ROS production, ultimately leading to apoptosis in embryonal carcinoma P19 cells [97] and zebrafish cells [98]. Moreover, given that the deposition of αSyn protein has been discovered in the neurons of stroke patients, and that FABP3 plays a pivotal role in αSyn oligomerization [45], it would appear very likely that the induction of αSyn neurotoxicity is one of the injury-related mechanisms of FABP3. The toxicity of αSyn is manifested by persistently elevated cytoplasmic Ca^2+^ levels, disruption of mitochondrial function, endoplasmic reticulum stress, and synaptic dysfunction [99,100], and FABP3 also causes endoplasmic reticulum stress and limits protein synthesis via protein kinase RNA-like endoplasmic reticulum kinase (PERK)–eukaryotic Initiation Factor 2α (eIF2α) signaling [101]. Taken together, these findings tend to indicate that an increase in FABP3 levels can accelerate ischemic neuronal apoptosis, which is favorable to the development of CI/RI, and that mitochondria are potential targets of attack.

### 5.2. FABP4

Under normal circumstances, brain tissues do not express FABP4, and, consequently, FABP4-related research has for a long time been focused primarily on tissues or organs other than the brain. However, Liao et al. were the first to discover the presence of abundant FABP4 in activated resident microglia and infiltrating monocyte-derived macrophages in the brains of MCAO mice, whereas little is detected prior to MCAO surgery [93]. The infiltration of peripheral monocytes in response to BBB disruption is considered to be one of the primary cellular origins of FABP4, and elevated levels of FABP4 has been shown to promote the expression and release of MMP-9 by enhancing c-Jun N-terminal kinase (JNK)/c–Jun signaling, thereby resulting in the degradation of tight junction proteins and BBB leakage, thus eventually aggravating ischemic brain injury.

In addition to the disruption of the BBB, FABP4 is also actively involved in apoptotic signaling activation, endoplasmic reticulum stress, oxidative stress, inflammation, and mitochondrial damage, all of which are important processes in the ischemic cascade. For example, FABP4 has been demonstrated to exacerbate I/R injury in mouse kidney tissue by modulating glucose-regulated protein 78 (GRP78)/C/EBP homologous protein (CHOP)/caspase-12 signaling to induce ER stress [92], whereas in a cellular I/R model, the involvement of FABP4 in endoplasmic reticulum stress has been found to be mediated via the PPARγ [102] and PI3K/Akt [103] pathways. As an important mediator of the inflammatory response, FABP4 is known to promote inflammation via mitochondrial uncoupling protein 2 (UCP2) [104,105], Toll-like receptor 4 (TLR4), and nuclear factor-κB (NF-κB) signaling [106,107]. Furthermore, following myocardial I/R, FABP4 knockout mice have been observed to have lower levels of myocardial infarct volume, inflammation, and oxidative and nitrative stress than wild-type mice [88], whereas the FABP4 inhibitor BMS309403 has been found to restore palmitic acid-induced mitochondrial dysfunction, including impaired respiratory complex IV and succinate dehydrogenase activity, and reduced mitochondrial membrane potential [108]. In conclusion, FABP4 appears to function as a generalist binding protein that is actively implicated in numerous cell injury pathways. However, it appears to induce ischemic neuronal apoptosis only indirectly via microglia and macrophages.

### 5.3. FABP5

FABP5 plays prominent roles in mitochondrial damage and the inflammatory response in cell injury. FABP5 aggregates in mitochondria in response to the administration of rotenone, a mitochondrial respiratory chain complex I inhibitor, thereby inducing oxidative damage in Neuro2A cells, participating in the formation of αSyn oligomers and exacerbating the further loss of mitochondrial membrane potential [52]. Moreover, in oligodendrocytes characterized by psychosine cytotoxicity, FABP5 interacts to VDAC-1 and BAX to form macropores in the mitochondrial membrane, releasing mitochondrial DNA and cytochrome *C* into the cytoplasm, thereby activating apoptotic caspases [53]. Consequently, elevated levels of FABP5 in the ischemic neurons of MCAO mice are likely to contribute directly to neuronal mortality by exacerbating mitochondrial damage. In addition, FABP5 play an important role as a mediator of the inflammatory effect of the pro-inflammatory cytokine Interleukin-1β (IL-1β), the levels of which are substantially elevated during ischemic injury. This can be attributed to the fact that the activation of NF-κB by IL-1β and subsequent entry into the nucleus are dependent on the presence of FABP5 [109]. Furthermore, levels of prostaglandin E2 (PGE2), a lipid signaling molecule involved in pain and inflammation, are elevated in the brain of MCAO mice, and deletion of its upstream synthetase, microsomal prostaglandin E synthase-1 (mPGES-1) effectively inhibits PGE2 synthesis and reduces brain infarction [110], whereas FABP5 precisely regulates mPGES-1 expression by controlling the binding of NF-κB to the binding site in the mPGES-1 promoter [109].

### 5.4. FABP7

In the ischemic penumbra region of MCAO mice, FABP7 is expressed and elevated predominantly in GFAP-positive astrocytes [78]. Effective blocking of mPGES-1-PGE2 signaling in ischemic brain tissue using a novel FABP7 inhibitor provides evidence to indicate that similar to FABP5, FABP7 regulates neuroinflammation. The overexpression of FABP7 can activate the pro-inflammatory phenotype of astrocytes and impair the survival of co-cultured motor neurons [111], thereby implying that FABP7 may indirectly participate in ischemic neuronal apoptosis by enhancing the inflammatory response of astrocytes. Furthermore, FABP7 expressed in astrocytes has been found to affect dendritic morphology and excitatory synaptic function in cortical neurons [112]. However, despite the fact that FABP7 expression is upregulated in neuronal stem/progenitor cells after ischemia, transient ischemia that fails to result in hippocampal neuronal death does not appear to influence the process whereby FABP7 regulates neurogenesis [83]. In addition, FABP7 has also been established to have a protective effect on astrocytes, which can contribute to the formation of lipid droplets that counter hypoxia-induced oxidative stress [113].

## 6. Potential of FABPs as Novel Therapeutic Targets in the Treatment of Ischemic Stroke

Although FABPs play indispensable roles in maintaining the functional homeostasis of cells, their overexpression has been established to be detrimental from the perspective of neuroprotection in ischemic stroke. Therefore, it is desirable to clarify the involvement of FABPs in the pathological mechanism of ischemic stroke. As shown in Figure 2, FABP3 and FABP5 are overexpressed in ischemic neurons, mediate mitochondrial damage and apoptotic protein activation under oxidative stress, and are directly involved in ischemic neuronal death. FABP4, localized in microglia, upregulates the expression and release of pro-inflammatory cytokines and matrix metalloproteinases, thereby activating apoptotic signaling in ischemic neurons and disrupting the BBB, whereas FABP7 activates a shift in resting astrocytes to a pro-inflammatory phenotype, and influences ischemic neuronal apoptosis indirectly via inflammation. In addition to FABP3, which is expressed exclusively in neurons, FABP5 is also expressed in oligodendrocytes, astrocytes, and vascular endothelial cells, and FABP7 is also found in oligodendrocytes. However, we have previously detected no significant changes in the expression of FABPs in these cells in response to ischemic events [78]. Within the neurovascular unit, these four FABP isoforms are expressed in different cell types, collectively contributing to the overall microcirculatory dysfunction of the brain. Given this organization, it is convenient to integrate different cell populations in the brain as a whole to study the processes of cerebral ischemic damage and repair, and, in recent years, the neurovascular unit has emerged as a new paradigm in the clinical treatment of ischemic stroke [23]. The presence of FABPs is intended to raise the single neuron protection, reperfusion therapy or other programs to new heights of holistic therapy, thereby compensating for the shortcomings of previous studies that have failed to take into account the integrity of brain function and the interaction between different structures.

The findings of research conducted in recent years have provided convincing evidence to indicate that pharmacological disruption of FABPs activity could be used as a novel strategy in the treatment of ischemic injury. For example, BMS309403, an inhibitor of FABP4, has been discovered to effectively ameliorate I/R injury in the brain [93] and kidneys [92] of mice. Similarly, we have previously developed a series of novel inhibitors for FABPs [25] and confirmed their neuroprotective effects. For example, we found that the FABP7 inhibitor MF6 can be applied to effectively reduce the cerebral infarct volume and neurological deficit in MCAO mice by inhibiting inflammation-related mPGES-1-PGE2 signaling [78]. However, we speculated that the pharmacological target of this inhibitor would not be confined exclusively to FABP7, given that in addition to having excellent affinity for FABP7 (dissociation constant (Kd) value: 20 ± 9 nM), MF6 also has high affinities for FABP3 (Kd value: 1038 ± 155 nM), FABP4 (530 ± 154 nM), and FABP5 (874 ± 66 nM) [25], and significantly reduces the protein expression of FABPs in the brain tissue [78]. On the basis of these findings, we plan to further investigate enhancing the inhibitory mechanism of MF6 and characterizing the FABP-mediated injury signals. Furthermore, it has been established that identified FABP inhibitors also have favorable therapeutic effects on other neurological diseases. For example, MF1 [114] and MF8 [115] have been demonstrated to target FABP3 and ameliorate motor deficits and cognitive impairment in MPTP-induced PD mouse models, the former of which enhances GABAA receptor activity via a benzodiazepine recognition site, thereby exhibiting anti-convulsant-like effects in pilocarpine-treated mice [116]. In summary, we are dedicated to examining the pathological role of FABPs in neurodegenerative diseases such as ischemic stroke and investigating the potential utility of FABPs inhibitors as neuroprotective agents.

## 7. Conclusions

Although in recent years there has been a large number of studies that have examined the roles of FABPs in diverse diseases, there is currently little information pertaining to the function of FABPs in the pathological processes of ischemic stroke. The mechanisms underlying ischemic stroke injury are complex and interrelated, and probably involve multiple interacting processes. However, as a family of lipid chaperone proteins that play multiple cell regulatory roles, FABPs are assumed to play pivotal roles in the occurrence of ischemic stroke associated with lipid abnormalities. In this review, we discuss the links between FABPs and ischemic stroke and summarize the trends and molecular mechanisms of FABPs. Among the different FABP isoforms, FABP3, FABP4, FABP5, and FABP7 have been established to be functional in different cell types in the neurovascular unit of the brain, in which they actively respond to the ischemia cascade and collectively aggravate the ischemic damage of brain tissue. Consequently, we believe that FABPs meet the requirements of multi-effect targets that can be regarded as potential pharmacological targets for neuroprotective therapy in ischemic stroke. Importantly, the discovery of pharmacological interventions targeting the neurovascular unit as a whole is considered an important novel approach in the treatment of this disease.

## Figures and Tables

**Figure 1 ijms-23-09648-f001:**
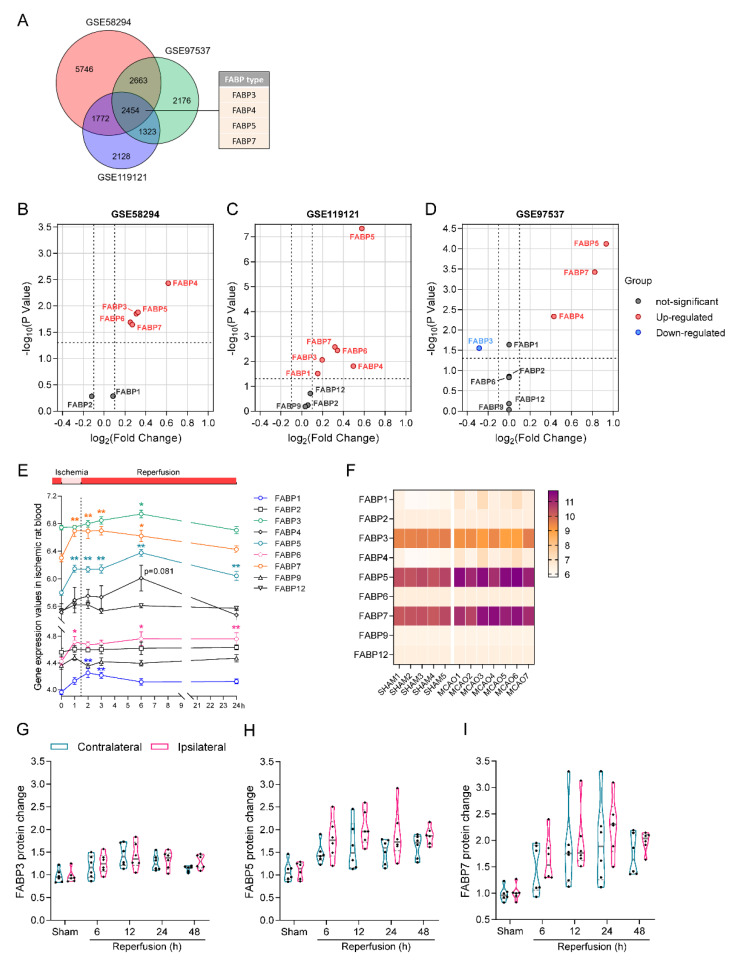
Differentially expressed genes (DEGs) induced by ischemic stroke. (**A**) A Venn diagram showing the overlap of DEGs among blood samples from ischemic stroke patients (GSE58294, 24 h after stroke), blood samples from MCAO rats (GSE119121, 6 h after MCAO), and brain tissue samples from MCAO rats (GSE97537, 24 h after the reperfusion). (**B**–**D**) Volcano plots showing *p*-values and log2 fold changes of gene expression for the three RNA-seq datasets in (**A**). Significant genes at *p* < 0.05, significantly up-regulated genes (Red dots), significantly down-regulated genes (blue dots) and non-significant genes (black dots). (**E**) FABP mRNA in whole blood was measured at 0, 1, 2, 3, 6 and 24 h after the onset of cerebral ischemia. (*n* = 8 per group, GSE119121). The data shown in each case represent the mean ± SEM. * *p* < 0.05, ** *p* < 0.01 vs. 0 h. (*n* = 8 per group) (**F**) Heatmap showing FABP gene expression levels in the brain of MCAO and sham rats 24 h after ischemia–reperfusion (GSE97537). (**G**–**I**) Mice received right cerebral ischemia for 2 h. Protein levels of FABPs in ischemic ipsilateral and contralateral non-ischemic brain tissue at 6, 12, 24 and 48 h after reperfusion. (*n* = 6 per group).

**Figure 2 ijms-23-09648-f002:**
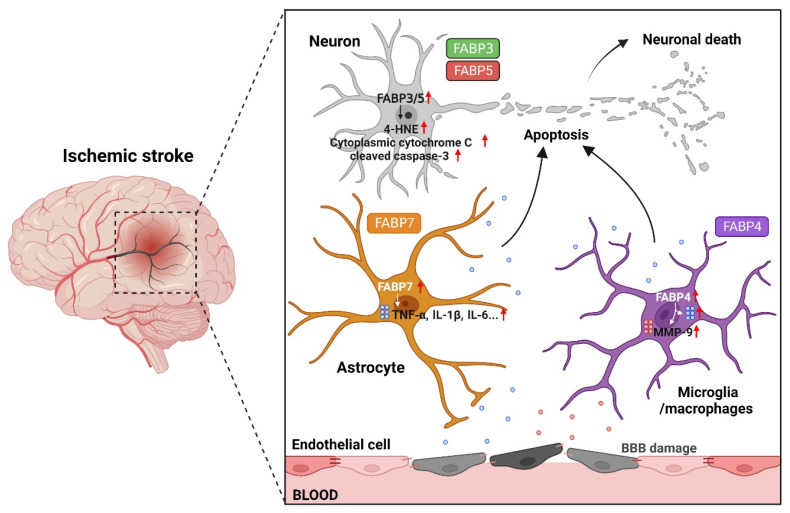
A schematic diagram of the pathway of Faty acid-binding proteins (FABPs) involved in the ischemic injury of neurovascular units. FABP3 and FABP5 are overexpressed in ischemic neurons, mediate oxidative stress, and directly participate in ischemic neuronal death. FABP4 expressed in microglia stimulates the expression and release of pro-inflammatory cytokines (IL-1β, IL-6, TNF-α), and matrix metalloproteinase MMP-9 activates extrinsic apoptotic signals in ischemic neurons and disrupts the blood–brain barrier (BBB). FABP7 enhances the pro-inflammatory expression of astrocytes and stimulates ischemic neuronal apoptosis through inflammation. The figure was created with BioRender.com.

**Table 1 ijms-23-09648-t001:** Changes in fatty acid-binding proteins following diverse ischemic episodes and their regulatory mechanisms.

Name	Aliases	Sample Source	Pathological State	Alteration	Pathological Functions	References
FABP1	Liver FABP (L-FABP)	Urine	Human, renal transplantation I/R	Up	—	[64]
		Urine	Mice, renal I/R model (30 min)		Elimination of lipid peroxidative products	[64,69]
					(4-HNE, 4-HHE)	
FABP2	Intestine FABP (I-FABP)	Serum	Human, acute intestinal ischemia	Up	—	[65,66]
		Serum, plasma	Rats, mesenteric ischemia model		—	[70,71,72,73]
		Urine	Human, acute mesenteric ischemia		—	[74]
FABP3	Heart FABP (H-FABP)	Serum	Human, acute ischemic stroke	Up	—	[67]
		Plasma	Human, ischemic cardiomyopathy		—	[75]
		Plasma	Human, acute myocardial infarction		—	[76,77,78]
		Serum	Human, aneurysmal subarachnoid hemorrhage		—	[78]
		Brain	Mice, MCAO model (2 h)		—	[79]
		Heart, serum	Mice, myocardial infarction model		Pro-apoptosis; ↑ MAPK phosphorylation, ↓ AKT phosphorylation	[68]
FABP4	Adipose FABP (A-FABP)	Serum	Human, acute ischemic stroke	Up	—	[80,81,82]
		Heart	Mice, myocardial I/R model (30 min)		↑ O_2_^•^^−^ and ONOO-; ↑ TNF-α, MCP-1 and IL-6	[83]
		Kidney	Mice, renal I/R model (30 min)		Pro-apoptosis; ↑ ER stress	[84]
		Liver	Mice, liver I/R model (60 min)		↑ IL-1β, IL-6, and TNFα	[85]
		Cortex, serum	Mice, MCAO model (1 h)		↑ JNK/c-Jun signaling, ↑ MMP-9	[86]
FABP5	Epidermal FABP (E-FABP)	Lung	Mice, PH-LHD (myocardial infarction) model	Up	↑ Wnt/β-catenin pathway	[87]
		Kidney	Pig, renal I/R model (30 min)		—	[88]
		Hippocampus	Monkeys, global cerebral I/R model (20 min)		—	[89]
		Brain	Mice, MCAO model (2 h)		—	[79]
FABP6	Ileal FABP (IL-FABP)	Plasma	Rats, Hemorrhagic shock model	Up	—	[90]
FABP7	Brain FABP (B-FABP)	Serum	Human, acute ischemic stroke	Up	—	[67]
		Hippocampus	Mice, BCCAO model (20 min)		↑ Neurogenesis	[91]
		Cortex	Monkeys, global cerebral I/R model (20 min)		—	[92]
		Hippocampus	Monkeys, global cerebral I/R model (20 min)		—	[89,93]
		Brain	Mice, MCAO model (2 h)		↑ Inflammation-associated mPGES-1, PGE2	[79]
FABP8	Myelin FABP (M-FABP)	—	—	—	—	—
FABP9	Testis FABP (T-FABP)	—	—	—	—	—
FABP12	Retinal-FABP (R-FABP)	—	—	—	—	—

“Human” represents clinical research data of patients, while preclinical data refers to animal “model” experiments in the table. “—” means no data available. “↑” means up-regulation and “↓” means down-regulation. MCAO: Middle cerebral artery occlusion; BCCAO: Bilateral common carotid artery occlusion; 4-HNE: I/R: Ischemia/Reperfusion; PH-LHD: Pulmonary hypertension-left heart disease; 4-HNE: 4-Hydroxynonenal; 4-HHE: 4-Hydroxyhexenal; MAPK: Mitogen-activated protein kinase; AKT: Protein kinase B; TNF-α: Tumor necrosis factor-α; MCP-1: Monocyte chemoattractant protein-1; IL-6: Interleukin 6; IL-1β: Interleukin-1β; JNK: c-Jun NH_2_-terminal kinases; MMP-9: Matrix metalloproteinase-9; mPGES-1: Microsomal prostaglandin E synthase-1; PGE2: Prostaglandin E2.

## Data Availability

The data that support the findings of this study are available from the corresponding author [Q.G.] upon reasonable request.

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
