# Peer review of "Fatty Acid-Binding Proteins: Their Roles in Ischemic Stroke and Potential as Drug Targets"

_ijms, 2022, doi:10.3390/ijms23179648_

Round 1

Reviewer 1 Report

The authors demonstrated the effects of fatty acid-binding proteins in processes following ischemic stroke. The different types of FA-binding proteins, their characteristics and importance in physiological and pathological conditions with special emphasis on stroke are described in detail.   The paper is coherent and clear, each paragraph is thoroughly described and supported by relevant citations.

Minor comments:

Lines:

 16 – double “this”

54  and 223 – the same title of paragraph

55 and 65 – please use one abbreviation

175-178 (appriopriate citation is missing) and 351-353 - Are the data presented from the same source?

424-428 - appropriate citations are missing

Table 1 – I propose to improve the table a bit to make it clearer, perhaps it would be better to show data from clinical and preclinical studies separately, the last three items I suggest to remove. With data from clinical trials, it would also be useful to show links between FABPs concentrations and other symptoms presented by the authors of each study.

Fig 2 – I propose to improve the drawing, the inscriptions in the cells are poorly visible, you can highlight more details such as specific cytokines, larger arrows

Author Response

  1. Comment: [16 – double “this”]

Response: Thank you so much for your careful check. We deleted one "this" (line 16)

  1. Comment: [54 and 223 – the same title of paragraph]

Response: We have revised “4. Pathophysiology of ischemic stroke” to “4. Changes of FABPs in ischemic stroke” (line 343)

  1. Comment: [55 and 65 – please use one abbreviation]

Response: We have revised “Cerebral ischemia/reperfusion (I/R)” to “CI/RI” (line 73)

  1. Comment: [175-178 (appriopriate citation is missing) and 351-353 - Are the data presented from the same source?]

Response: We inserted references in the appropriate positions. “Abnormal accumulation of FABP5 …… Bcl-2-associated X protein (BAX) [53, 54]” (line 242-245)

                   Line 351-353: “Moreover, in oligodendrocytes characterized by psychosine cytotoxicity, FABP5 interacts to VDAC-1 and BAX to form macropores in the mitochondrial membrane, releasing mitochondrial DNA and cytochrome C into the cytoplasm, and thereby activating apoptotic caspases [53]” was presented from a research paper. (line 606-628)

  1. Comment: [424-428 - appropriate citations are missing]

Response: We added references in the appropriate positions. “However, we speculated ……, MF6 also has high affinities for FABP3 (Kd value: 1038 ± 155 nM), FABP4 (530 ± 154 nM), and FABP5 (874 ± 66 nM) [25], and significantly reduces the protein expression of FABPs in the brain tissue [78]” (line 711-716)

  1. Comment: [Table 1 – I propose to improve the table a bit to make it clearer, perhaps it would be better to show data from clinical and preclinical studies separately, the last three items I suggest to remove. With data from clinical trials, it would also be useful to show links between FABPs concentrations and other symptoms presented by the authors of each study.]

Response: We tried to modify Table.1 as you suggested, but given the layout and size of the table, we found that it seems difficult to further break down the clinical studies and preclinical studies in the table, and we believe that the “Human” and “Mice/Rats/Pig/Monkeys” in the table are sufficient to understand whether the data are clinical or preclinical studies. Moreover, we have added the word “model” to the preclinical data in the Table.1 and described it in the legend ““Human” represents clinical research data of patients, while preclinical data refers to animal “model” experiments in the table. “-” means no data available” (line 338). If you think it was not clear enough, please correct us again.

In addition, we considered your suggestion to delete the last three items, but we would prefer not to do so, because we thought that the current situation of not finding reports of FABP8, 9 and 12 associated with ischemic events is a result that can alert readers to the uncharted territory that can be explored. If you insist on removing them, please correct us again.

  1. Comment: [Fig 2 – I propose to improve the drawing, the inscriptions in the cells are poorly visible, you can highlight more details such as specific cytokines, larger arrows]

Response: Thanks again for your valuable suggestion, we modified Figure.2 according to your suggestion.

Reviewer 2 Report

The manuscript makes real contributions in the field of ischemic stroke. We recommend that the authors carefully revise the manuscript once more to correct spelling mistakes and repetitions.

Author Response

  1. Comment: [We recommend that the authors carefully revise the manuscript once more to correct spelling mistakes and repetitions]

Response: We have carefully revised the manuscript for spelling mistakes and repetitions.

Reviewer 3 Report

The authors have investigated dynamics of FABP in various brain diseases such as α-synucleinopathy and cerebral ischemia/reperfusion injury, and reviewed the literature published in the past decade that had reported on the associations between FABPs and cerebral ischemia, and summarized the relevant regulatory mechanisms of FABPs implicated in cerebral ischemic injury.

I think the manuscript is well written, but several points should be amended to be accepted in the journal.

#1. The authors used the term ‘cerebral ischemia/reperfusion injury’ in the manuscript. I would like to ask the authors which readers the manuscript is mainly for basic scientists or physicians. That is, cerebral ischemia/reperfusion injury in basic research can be induced by MCAO in mice and rats, while that in clinical settings is occasionally observed after carotid endoarterectomy and carotid artery stenting, which are limited occasions. The authors should carefully mention the term, and add explanations.

#2. The authors mentioned that FABP8 is found abundantly in the myelin sheaths and Schwann cells of peripheral nerves, and has been found to function in tandem with myelin basic protein to stack phospholipid membranes and exert unique functions in the organization and stabilization of myelin multilayers [23]. I would like to know whether elevation of FABP8 is associated with small vessel disease. Please add some explanation if published articles are found.

#3. 3.2. Functions of FABP3, 5, and 7 in neurodegeneration seems to be redundant. Please summarize the contents because main theme of the manuscript is with regard to ischemic stroke. I understand the importance of FABP and neurodegenerative diseases.

#4. In Figure 1, the authors should add explanation when mRNA was extracted after MCAO (e.g., after 2 hours, 1 day….), and when blood was collected in patients with cardioembolic stroke.

#5. It is unclear why the dissociation of FABP3 between human blood and rat brain was observed in Figure 1.

Author Response

  1. Comment: [The authors used the term ‘cerebral ischemia/reperfusion injury’ in the manuscript. I would like to ask the authors which readers the manuscript is mainly for basic scientists or physicians. That is, cerebral ischemia/reperfusion injury in basic research can be induced by MCAO in mice and rats, while that in clinical settings is occasionally observed after carotid endoarterectomy and carotid artery stenting, which are limited occasions. The authors should carefully mention the term, and add explanations.]

Response: We included a more detailed explanation to describe the term "cerebral ischemia/reperfusion injury" as follows:

   We added "which are sources of additional damage to the cerebral microcirculation and adjacent brain tissue, a condition known as cerebral ischemia/reperfusion injury (CI/RI) [3, 6] " (line 43-44)

   We added "CI/RI is defined as a damage caused by the rapid restoration of blood supply to brain tissue after a period of ischemia, and is a common feature of ischemic stroke [11]. It is thought to play a major role in the pathophysiology of acute ischemic stroke, be-cause secondary brain injury owing to abrupt reperfusion has been shown in preclini-cal studies to contribute to 70% of the final ischemic lesions [12]. Spontaneous reperfu-sion occurs in approximately 50-70% of ischemic stroke patients [13]. There is no doubt that CI/RI has become a serious problem, leading to a poor prognosis in stroke patients [14, 15]" (line 57-63)

  1. Comment: [The authors mentioned that FABP8 is found abundantly in the myelin sheaths and Schwann cells of peripheral nerves, and has been found to function in tandem with myelin basic protein to stack phospholipid membranes and exert unique functions in the organization and stabilization of myelin multilayers [23]. I would like to know whether elevation of FABP8 is associated with small vessel disease. Please add some explanation if published articles are found.]

Response: We are sorry that we cannot explain whether elevation of FABP8 is associated with small vessel disease, because we have not found any relevant reports. Currently, there have been limited studies on the functions of FABP8 in both normal physiological and pathological states, and these studies only support that FABP8 is significantly expressed in the sciatic nerve endoneurium of mice 2 weeks after birth and regulates lipid metabolism in medullary Shewan cells [1], and that several point mutations in FABP8 have been associated with the hereditary neuropathy known as Charcot-Marie-Tooth disease [2, 3]. Thank you for raising this issue and we will continue to pay attention to this issue in the future.

[1] Zenker J, Stettner M, Ruskamo S, et al. A role of peripheral myelin protein 2 in lipid homeostasis of myelinating Schwann cells. Glia. 2014;62(9):1502-1512. doi:10.1002/glia.22696

[2] Motley WW, Palaima P, Yum SW, et al. De novo PMP2 mutations in families with type 1 Charcot-Marie-Tooth disease. Brain. 2016;139(Pt 6):1649-1656. doi:10.1093/brain/aww055

[3] Punetha J, Mackay-Loder L, Harel T, Coban-Akdemir Z, Jhangiani SN, Gibbs RA, Lee I, Terespolsky D, Lupski JR, Posey JE. Identification of a pathogenic PMP2 variant in a multi-generational family with CMT type 1: Clinical gene panels versus genome-wide approaches to molecular diagnosis. Mol Genet Metab. 2018 Nov;125(3):302-304. doi: 10.1016/j.ymgme.2018.08.005.

  1. Comment: [3.2. Functions of FABP3, 5, and 7 in neurodegeneration seems to be redundant. Please summarize the contents because main theme of the manuscript is with regard to ischemic stroke. I understand the importance of FABP and neurodegenerative diseases.]

Response: The purpose of this manuscript is to review the role of FABPs in ischemic stroke and to present the idea of their potential as drug targets. Although all available evidence supports the close association of FABPs with ischemic brain injury, relevant basic studies are insufficient and there is still a lack of relevant reports on the specific mechanisms of FABPs involved in ischemic brain injury, so we tried to indirectly speculate on the potential regulatory roles of FABPs in ischemic stroke by introducing the mechanisms of FABPs in neural injury in other neurodegenerative diseases. Currently, most of the FABPs closely associated with neurodegenerative diseases are focused on FABP3, FABP5 and FABP7, so we present the functions of these three FABPs in neurodegeneration for the readers’ reference. If you insist this is redundant, please correct us again.

  1. Comment: [In Figure 1, the authors should add explanation when mRNA was extracted after MCAO (e.g., after 2 hours, 1 day….), and when blood was collected in patients with cardioembolic stroke.]

Response: Thank you for the valuable reminder. We have revised the legend of Figure 1.

" Figure 1. Differentially expressed genes (DEGs) induced by ischemic stroke. (A) A Venn diagram showing the overlap of DEGs among blood samples from ischemic stroke patients (GSE58294, 24h after stroke), blood samples from MCAO rats (GSE119121, 6h after MCAO), and brain tissue samples from MCAO rats (GSE97537, 24h after the reperfusion). (B-D) Volcano plots showing p-values and log2 fold changes of gene expression for the three RNA-seq datasets in (A). " (line 481-485)

  1. Comment: [It is unclear why the dissociation of FABP3 between human blood and rat brain was observed in Figure 1.]

Response: Our understanding of your question is why the changes in the blood of patients with FABP3 do not match the changes in the MCAO rat brains. We thought that the duration of ischemia and the region of brain tissue collected may be the main reason why the changes in FABP3 in MCAO rats are different from other datasets. This dataset does not explicitly indicate the duration of the ischemic state, and different severities of brain injury have a significant impact on the protein expression. Moreover, ischemic events result in ischemic core and penumbra regions in the brain, and in our previous study, we observed that FABP3 was significantly expressed in neurons in the ischemic penumbra, whereas FABP3 was degraded or reduced in the core regions due to neuronal necrosis/fragmentation, but there is no detailed description of the collected brain tissues on this database page (https://www.ncbi.nlm.nih.gov/geo/query/acc.cgi?acc=GSM2571735). Furthermore, it cannot be excluded that elevated FABP3 in the blood of MCAO rats is outflow from the ischemic core due to BBB disruption. We regret that there is not enough evidence to explain the occurrence of this phenomenon.